# Flutamide Promotes Early Hepatocarcinogenesis Through Mitophagy in High-Fat Diet-Fed Non-Obese Steatotic Rats

**DOI:** 10.3390/ijms26062709

**Published:** 2025-03-17

**Authors:** Emika Hara, Kanami Ohshima, Mio Takimoto, Yidan Bai, Mai Hirata, Wen Zeng, Suzuka Uomoto, Mai Todoroki, Mio Kobayashi, Takuma Kozono, Tetsuhito Kigata, Makoto Shibutani, Toshinori Yoshida

**Affiliations:** 1Laboratory of Veterinary Pathology, Cooperative Department of Veterinary Medicine, Tokyo University of Agriculture and Technology, 3-5-8 Saiwai-cho, Fuchu-shi, Tokyo 183-8509, Japanmshibuta@cc.tuat.ac.jp (M.S.); 2Cooperative Division of Veterinary Sciences, Tokyo University of Agriculture and Technology, 3-5-8 Saiwai-cho, Fuchu-shi, Tokyo 183-8509, Japan; 3Smart-Core-Facility Promotion Organization, Tokyo University of Agriculture and Technology, 3-5-8 Saiwai-cho, Fuchu-shi, Tokyo 183-8509, Japan; tkozono@go.tuat.ac.jp; 4Laboratory of Veterinary Anatomy, Cooperative Department of Veterinary Medicine, Tokyo University of Agriculture and Technology, 3-5-8 Saiwai-cho, Fuchu-shi, Tokyo 183-8509, Japan; fq6451@go.tuat.ac.jp

**Keywords:** AMBRA1, flutamide, LC3, mitophagy, NAFLD

## Abstract

Flutamide (FL), a non-steroidal drug used for its antiandrogenic, anticancer, and disrupting endocrine properties, induces mitochondrial toxicity and drug metabolism enzymes and promotes hepatocarcinogenesis. The inhibition of mitophagy, leading to the accumulation of damaged mitochondria, is implicated in the pathogenesis of nonalcoholic fatty liver disease (NAFLD). In this study, we investigated the effects of FL in high-fat diet (HFD)-induced non-obese steatosis rats, categorized into four groups: basal diet (BD), BD + FL, HFD, and HFD + FL. The FL exacerbated HFD-induced steatosis and marginally increased preneoplastic lesions. To analyze hepatic preneoplastic lesions, we divided them into clusters based on the expression ratios of the mitophagy regulators LC3 and AMBRA1. The expression rates of LC3 and AMBRA1 in these precancerous lesions were classified into three clusters using *k*-means clustering. The HFD group exhibited an increased ratio of mitophagy inhibition clusters, as indicated by decreased LC3 and increased AMBRA1 levels in background hepatocytes and preneoplastic lesions. FL counteracted HFD-mediated mitophagy inhibition, as indicated by increased LC3 and decreased AMBRA1 levels in background hepatocytes. Our clustering analysis revealed that FL-induced mitophagy induction relied on Parkin expression. The present study underscores the significance of cluster analysis in understanding the role of mitophagy within small preneoplastic lesions and suggests that FL may potentially exacerbate NAFLD-associated hepatocarcinogenesis by affecting mitophagy.

## 1. Introduction

Nonalcoholic fatty liver disease (NAFLD) is a severe liver disease, globally affecting approximately 25% of the population and imposing a significant health burden along with widespread social and economic consequences [1]. Metabolic dysfunction-related fatty liver disease (MAFLD) has been proposed instead of NAFLD, because it is associated with metabolic abnormalities. Increased fatty acid oxidation within the mitochondria plays a crucial role in preventing fat accumulation [2]; however, a chronic high-fat diet (HFD), excessive glucose intake, and genetic obesity lead to increased oxidation rates, resulting in increased reactive oxygen species (ROS) generation [3]. Moreover, reduced hepatic ATP synthesis, lowered mitochondrial respiratory chain complex activity, and abnormal mitochondria have all been reported in NAFLD patients [4,5,6]. Furthermore, mitochondrial dysfunction promotes abnormal hepatic lipid homeostasis, insulin resistance, cell death, and cytokine release, thereby leading to progression from NAFLD to nonalcoholic steatohepatitis (NASH) and eventually cirrhosis [7]. Some patients also have the risk of developing liver cancer with or without cirrhosis [8]. Therefore, promoting the removal of dysfunctional mitochondria, which contributes to fatty liver development, may play a pivotal role in the treatment of NAFLD.

Autophagy is a catabolic process that eliminates inessential intracellular materials through lysosomal degradation, thereby supplying nutrients and energy for cellular homeostasis [9]. Cells often activate autophagy to cope with various stressors, such as nutrient scarcity, protein misfolding, and damaged organelles. Autophagy serves to clear selectively lipid droplets and abnormal organelles [10]. The selective process, known as mitophagy, focuses on the removal of impaired mitochondria and has been extensively studied over the past decade [11]. Mitophagy involves two steps: induction of general autophagy and priming of damaged mitochondria for selective autophagy. Recent advances in mitophagy research have revealed that mitochondrial priming is mediated by phosphatase and tensin homolog (PTEN)-induced kinase 1 (PINK1)-Parkin-dependent or -independent mechanisms [11,12,13]. These targeted mitochondria are incorporated into the sequestration membrane (phagophore) by the microtubule-associated protein 1A/1B-light chain 3 (LC3) and mitophagy-related receptors, which bind to the phagophore to form autophagosomes. Subsequently, these autophagosomes fuse with lysosomes to form autolysosomes, where damaged mitochondria are degraded. In this process, the autophagy and Beclin 1 regulator 1 (AMBRA1) are pivotal as mitophagy regulators involved in both Parkin-dependent and -independent pathways [14]. In the Parkin-dependent mitophagy, activated Parkin interacts with AMBRA1 expressed on the outer mitochondrial membrane, which, in turn, promotes mitophagy by activating the class III phosphatidylinositol 3-kinase (PI3K) complex, an essential contributor to autophagosome formation [15,16]. Additionally, AMBRA1 directly binds to LC3, further driving the progression of autophagosome formation [17].

Accumulating evidence suggests that the inhibition of mitophagy increases with the progression of NAFLD. In a murine model of HFD-induced NAFLD, autophagy was arrested, leading to mitochondrial dysfunction in the hepatocytes [18]. In a similar model, the progression of mitophagy was suppressed, evident in the accumulation of megamitochondria containing mitophagy intermediates such as p62/SQSTM1 and ubiquitin [19]. In primary hepatocytes from mice treated with palmitic acid to mimic high-fat stress, the expression levels of LC3 and Parkin in the mitochondria were notably reduced, indicating that mitophagy is inactivated in a Parkin-dependent manner [20]. PINK1-Parkin-dependent mitophagy is inhibited in the livers of NAFLD models and patients with NAFLD [21].

Flutamide (FL) is a non-steroidal antiandrogenic and anticancer drug widely used to treat prostate cancer and recognized as an endocrine-disrupting environmental contaminant [22,23,24]. Moreover, FL is a mitochondrial toxicant, inhibiting mitochondrial respiratory chain complexes I, II, and V [25,26]. In a previous study conducted in our laboratory, FL was observed to enhance precancerous lesions in a medium-term hepatocarcinogenesis model in rats; however, the underlying mechanism remains unclear [27]. In this study, we hypothesized that FL induces mitochondrial injury followed by mitophagy in an HFD-mediated early hepatocarcinogenesis model, which served as valuable models for understanding the augmentation of autophagy in precancerous lesions [28,29,30,31], according to an initiation and promotion model [32]. We can identify hepatic precancerous lesions expressing the autophagosome marker LC3 and the cargo receptor p62 [29,31,33]. We investigated the role of mitophagy in NAFLD-related hepatocarcinogenesis by clustering hepatic preneoplastic lesions according to the expression ratios of LC3, AMBRA1, and Parkin, based on a previously established method [34].

## 2. Results

### 2.1. Flutamide Inhibits Body Weight Growth, Increases HFD-Induced NAFLD Score, and Enhances Preneoplastic Hepatic Lesions

During the study period, FL suppressed body weight gain after 6 weeks, regardless of the diet type (Figure 1a; Appendix A). No discernible difference was observed in food and water intake (Table 1, Appendix A). In the final autopsy, FL significantly reduced the final body weight and intra-abdominal fat weight, independent of the HFD (Table 1). Importantly, HFD-induced obesity was not observed under these experimental conditions (Figure 1a; Table 1).

Previous studies have shown that an HFD increases NAS and steatosis scores in the liver [28,29,30,31]. Similar results were obtained in the present study, where FL further increased these scores (Figure 1b–d). Notably, the scores associated with ballooning degeneration and inflammatory foci increased in response to an HFD but remained unaffected by FL (Figure 1e,f). FL caused a significant increase in relative liver weight in the BD + FL group, which might have been caused by CYP induction [27], as evident from the *Cyp1a* gene expression data presented below (Table 2). However, no statistically significant change in relative liver weight was observed in the HFD + FL group (Table 1). We also evaluated an antiandrogenic effect of FL in reproductive organs [22,23,24]. The absolute and relative weights of epididymides, seminal vesicles/coagulating glands, and prostate, but not testis were significantly decreased in the FL groups (BD + FL, HFD + FL), compared with no-FL-treated groups (BD and HFD) (Table 1).

The consumption of HFD induced a significant increase in the number and area of glutathione S-transferase placental form (GST-P)-positive foci (Figure 1g–k), as previously reported [28,29,30,31]. A previous study showed that higher FL doses (1000 and 2000 ppm) enhanced hepatocarcinogenesis in rats [27]; however, in the present study, a lower FL dose (500 ppm) caused a marginal increase in both the number and area of GST-P-positive foci (Figure 1g–j). The area of GST-P-positive foci was significantly increased in the HFD + FL group compared to the BD group; however, the cumulative effects of HFD and FL on the area of GST-P-positive foci were marginal (Figure 1g–j). The number of Ki-67-positive cells within hepatic precancerous lesions showed an increasing trend in the FL-treated groups (BD + FL and HFD + FL), although no statistically significant change was detected (Appendix A). These findings suggest that FL exacerbated steatosis and marginally increased preneoplastic lesions in a rat model of non-obesity-related steatosis.

### 2.2. Expression Analysis of Mitophagy Markers in Background Hepatocytes and Precancerous Lesions

FL exhibits mitochondrial toxicity [35], while HFD inhibits mitophagy [21,36]. In light of these insights, we assessed the expression of the autophagy marker LC3 and the mitophagy indicator AMBRA1 in both background hepatocytes and precancerous lesions of rats treated with HFD and/or FL. LC3 and AMBRA1 showed positive granular signals in hepatocytes (Appendix A). Quantitative analysis showed that LC3-positive granules exhibited an increasing trend in the FL-treated groups (BD + FL and HFD + FL groups) compared to the non-FL-treated groups (BD and HFD groups) (Appendix A), both in background hepatocytes (non-foci) and precancerous lesions (foci). Similarly, AMBRA1-positive granules tended to increase in number in the HFD group compared to the BD group. However, when combined with FL, a decreasing trend was observed, particularly in the BD + FL group, in both background hepatocytes (non-foci) and precancerous lesions (foci) (Appendix A). These results indicate a potential suppression of mitophagy in the HFD group, as indicated by the increased expression of AMBRA1. Conversely, in combination with the FL treatment, mitophagy might be induced by the suppression of increased AMBRA1 expression; however, these effects on mitophagy were found to be marginal in this standard analysis.

No statistically significant differences in Parkin expression were observed in either background hepatocytes or preneoplastic lesions in each group (Appendix A).

### 2.3. Cluster Analysis of LC3, AMBRA1, and Parkin Expression in Background Hepatocytes

To further investigate the relationship between LC3 and AMBRA1 expression, we compared the expression ratios of both indicators in background hepatocytes using cluster analysis, as previously reported [34], aiming to elucidate the dynamics of autophagy flux (Appendix A). Using the *k*-means, the clusters were divided into three groups (Figure 2a–c) and the cluster composition of each group was calculated (Figure 2d). These clusters were classified into three levels: C1, characterized by low expression of both LC3 and AMBRA1, indicative of a basal level; C2, characterized by low AMBRA1 expression and high LC3 expression, suggestive of mitophagy induction; and C3, characterized by low LC3 expression and high AMBRA1 expression, implying mitophagy inhibition (Figure 2b). In the BD group, the major constitutive cluster was C1, followed by C3, and lastly C2 in descending order (Figure 2d). This pattern closely resembled that of the BD + FL group, where the primary constitutive cluster remained C1, similar to the BD group. However, the percentage of C2 clusters in the BD + FL group increased compared to the BD group, although this difference was not statistically significant. In contrast, in the HFD group, the prevalence of C3 composition significantly increased. In the HFD + FL group, however, the cluster composition was similar to that observed in the BD + FL group, with a significant increase in the rate of C2 composition.

We analyzed LC3 and Parkin expression to evaluate whether mitophagy induction was Parkin-dependent in the FL-treated groups (Figure 2e–h). The clusters were classified into four levels: C1, characterized by low LC3 expression and high Parkin expression, implying mitophagy inhibition; C2, with low Parkin expression and high LC3 expression, suggestive of Parkin-independent mitophagy induction; C3, a basal cluster with low expression of both LC3 and Parkin; and C4, featuring high expression of both LC3 and Parkin, indicative of Parkin-dependent mitophagy induction (Figure 2f). In the BD + FL group, the primary cluster was C3, while in the HFD + FL group, the C4 composition was significantly augmented (Figure 2h), suggesting that FL induced mitophagy in a Parkin-dependent manner in a population of preneoplastic lesions in HFD-fed rats.

### 2.4. Cluster Analysis of the GST-P-Positive Foci Area and LC3, AMBRA1, and Parkin Expression in Hepatic Precancerous Lesions

To determine whether AMBRA1 expression played a role in the development of hepatic preneoplastic lesions, we examined the relationship between the area of individual GST-P-positive foci and the expression rate of AMBRA1 in each lesion (Figure 3a–d). Subsequently, we classified the cluster into four levels, namely C1 to C4, and observed that C4 exhibited the highest AMBRA1 expression (mitophagy inhibition cluster) among all four levels (Figure 3b). The percentage of C4 composition significantly increased in the HFD group, and this effect was attenuated when combined with FL (Figure 3d), suggesting that FL might cancel mitophagy inhibition in HFD-fed rats.

We investigated the relationship between LC3 and AMBRA1 expression in hepatic precancerous lesions, which were divided into three clusters (Figure 3e–h), and examined the cluster composition of each group (Figure 3h). These clusters were classified into three levels, namely C1–C3: C1 represented the basal level with low expression of both LC3 and AMBRA1; C2 indicated mitophagy induction with low expression of AMBRA1 and high expression of LC3; and C3 denoted mitophagy inhibition with high expression of AMBRA1 and low expression of LC3 (Figure 3f), similar to the categorization in background hepatocytes (C3 in Figure 2b). The percentage of C3 was significantly higher in the HFD group than in the BD group (Figure 3h). The composition of clusters in the HFD + FL group was similar to that in the BD + FL group. The results suggested that FL might cancel mitophagy inhibition in HFD-fed rats as well.

Finally, we analyzed LC3 and Parkin expression in hepatic precancerous lesions to evaluate whether mitophagy induction was Parkin-dependent in the FL-treated groups (Figure 3i–l). The clusters were classified into three levels: C1 represented the basal level with low expression of both LC3 and Parkin; C2 indicated a cluster with low LC3 expression and high Parkin expression, indicative of mitophagy inhibition; and C3 represented a level with high LC3 expression and low Parkin expression, suggestive of Parkin-independent mitophagy induction (Figure 3j). Unlike in background hepatocytes, we did not detect C4, a Parkin-dependent mitophagy inhibition cluster (C4 in Figure 2f). C3 was minimally detected in the FL-treated groups (BD + FL and HFD + FL) (Figure 3l).

### 2.5. Flutamide Induces Autophagy

A transmission electron microscope (TEM) analysis was conducted in each group to observe steatosis and mitophagy in background hepatocytes (Appendix A). Fatty droplets were scattered in the HFD group. Autophagosomes, possibly mitophagy, were observed in FL groups (BD + FL and HFD + FL). Autolysosomes were noted in all the groups.

### 2.6. HFD and HFD Combined with FL Alters Hepatic Gene Expression

A comprehensive gene expression analysis was conducted on the liver samples to determine the contributions of autophagy, mitophagy, injured mitochondria, lipid metabolism, inflammation, drug-metabolizing enzymes, and oxidative stress-related genes, which might be related to NAFLD-related liver carcinogenesis.

We analyzed autophagy-related genes and showed that the expression of *Atg5*, *Atg7*, *Lamp1*, *Lamp2*, and *Lc3* was significantly increased in the HFD group compared to the BD group. However, when HFD feeding was combined with FL administration, a significant decrease was not observed in the expression of these genes except for *Lamp2* (Table 2). These gene expression changes in the HFD group were considered a compensatory response to the suppression of mitophagy, as discussed earlier. Regarding mitophagy-related genes, *Parkin* expression was significantly elevated in the HFD + FL group compared to the BD group, along with higher levels of *Ambra1* expression. Furthermore, a relatively higher increase in the expression of *rkin* and *Ambra1* was observed in the BD + FL and/or HFD groups. This could potentially indicate the presence of Parkin-dependent mitophagy in background hepatocytes, aligning with the observations discussed earlier.

FL, a known inhibitor of the mitochondrial respiratory chain complex I, has been reported to impair mitochondrial functions [25]. Moreover, 2-hydroxyflutamide, a major FL metabolite, impairs the functions of mitochondrial respiratory chain complexes II and V [26]. In this study, we analyzed the gene expression of complex I (*NADH*: quinone oxidoreductase), complex II (SDHD: succinate dehydrogenase), and complex V (*ATP synthase*). We found that the expression level of *Sdhd* tended to increase in the BD + FL group and was significantly elevated in the HFD group compared to the BD group. However, this expression trend reversed with the context of combined HFD feeding and FL administration, indicating a decreasing pattern (Table 2). Although no evident differences in the expression of *Nadh* and *ATP synthase* were observed between the groups, the expression of *Nadh* tended to increase in the HFD group; however, it remained comparable to that in the BD group. It is important to note that FL induces significant upregulation of *Cyp1a1* expression in rats [37]. In our study, the expression of *Cyp1a1 was* significantly elevated upon FL administration, regardless of the presence of HFD.

Regarding the expression levels of genes related to lipid metabolism, *Abca1*, which is responsible for the production of high-density lipoproteins, and *Srebf2*, which is involved in cholesterol metabolism, were significantly increased in the HFD groups (HFD and HFD + FL) compared to the BD group (Table 2). In addition, the TG synthase gene *Dgat2*, steroid hormone synthase *Hsd3b1*, and transcription factor *Ppara*, which promotes β-oxidation, were significantly increased in the HFD group compared to the BD group, while they either showed a declining trend or a significant decrease in the HFD + FL group.

No obvious differences in the expression of *Tnf-α*, one of the most effective cytokines, was seen between the groups (Table 2).

Regarding oxidative stress gene expression, the antioxidant enzyme *Catalase* exhibited a significant increase in the HFD group compared to the BD group. Furthermore, the expression level of *Gpx2* significantly increased in the FL administration alone and in the combination of HFD feeding and FL administration (Table 2). Thus, in the HFD group, antioxidant enzyme-related genes exhibited increased expression parallel to steatosis. However, the combined effect of HFD feeding and FL administration on gene expression was not detected.

## 3. Discussion

In recent years, the global incidence of NAFLD, also known as MAFLD, has been on the rise, and a subset of patients has developed liver cancer [8]. Mitophagy is impaired in patients with NAFLD, potentially contributing to the persistence of abnormal mitochondria and excessive ROS generation, which in turn may exacerbate the fatty liver [19,20,21]. However, the precise role of mitophagy in NAFLD-related hepatocarcinogenesis remains unclear. To examine the role of mitophagy in the non-obese rat model of NAFLD-related hepatocarcinogenesis, we chose to utilize FL as a mitochondrial toxicant. The effects of 500 ppm FL in inducing mitochondrial toxicity and hepatocarcinogenesis were found to be marginal, as evidenced by limited alterations in mitochondria/mitophagy-related gene expression (Table 2), and a gradual increase in the area of hepatic precancerous lesions compared to the BD group (Figure 1h–j). Consistent with a previous study with 1000 or 2000 ppm FL in rats [27], we observed a noticeable expression of *Cyp1a1* in the FL-treated liver (Table 2); however, CYP-mediated oxidative stress was not involved in hepatocarcinogenesis [27,38]. In this study, a significant increase in *Gpx2* gene expression in the BD + FL group suggested that FL induced an oxidative stress response; however, the effect remained limited, as other antioxidant genes were largely unaffected. The antiandrogenic effects of 500 ppm FL on reproductive organs were demonstrated entirely in the present study condition (Table 1). To challenge the elucidation of the mechanism, we assessed mitophagy flux using immunohistochemistry-based clustering analysis with mitophagy markers in background hepatocytes and preneoplastic lesions, following a previously reported method [34].

To understand the dynamics of mitophagy flux, it is essential to confirm the progression of mitophagy by identifying various autophagy markers. In selective autophagy, the initial reaction involves the formation of a phagophore that surrounds the cargo, ultimately giving rise to autophagosomes [9,10,36]. Subsequently, lysosomes merge with these autophagosomes to form autolysosomes, where lysosome-derived enzymes degrade the cargo, and the degraded products are reused within cells. This sequence of events constitutes the autophagic flux (Appendix A) [39,40]. In general, LC3 is expressed both inside and outside phagophores and autophagosomes, making it a widely used marker for examining autophagy flux [39,41]. In our study, LC3 was observed as granules in hepatocytes, suggesting the accurate detection of autophagosomes (Appendix A). LC3 includes two isoforms: LC3-I, which is ubiquitously expressed in autophagosomes, and LC3-II, which is increased in the progression of autophagy. It is noteworthy that the anti-LC3 antibody used in our study does not discriminate between LC3-I and LC3-II. Therefore, when analyzing autophagic flux, it becomes essential to observe specific molecules expressed on cargo receptors and injured intracellular organelles that are taken up by autophagosomes. Increases in the expression of cargo receptors and injured organelle markers indicate inhibition of autophagy [39,40]. We analyzed mitophagy flux by combining LC3 with the mitophagy regulatory factors AMBRA1 and Parkin [14,15,16,17,20,42,43], which are expressed on the membranes of impaired mitochondria (Appendix A).

In the livers of patients with NAFLD, the suppression of autophagy has been linked to increased accumulation of fat droplets, as determined by analyzing the expression of the autophagosome markers LC3-I/LC3-II and cargo receptor p62 [44,45]. These findings are recapitulated in in vitro models stimulating NAFLD liver conditions. In studies employing hepatocytes and hepatocellular carcinoma cells, the inhibition of mitophagy has been shown to increase fat droplet deposition, as indicated by the altered expression patterns of mitophagy indicators such as AMBRA1, Parkin, and PINK1 [20,21]. In the present study, although no distinct differences were observed in the standard analysis of LC3-, AMBRA-, and Parkin-positive reactions in background hepatocytes (Appendix A), our clustering analysis revealed that HFD increased the proportion of mitophagy inhibition clusters with high AMBRA1 expression (C3 in Figure 2b–d). The higher expression of *Atg5*, *Atg7*, *Lamp1*, *Lamp2*, and *Lc3* in the HFD group (Table 2) could potentially reflect an adaptive response to the processing of excess fat droplet load caused by mitophagy inhibition. In contrast, the HFD + FL group showed an increased proportion of clusters indicative of mitophagy induction (C2 in Figure 2b–d), a response dependent on Parkin (C4 in Figure 2f–h). Remarkably, the increased expression of *Atg5*, *Atg7*, *Lamp1*, *Lamp2*, and *Lc3* in the HFD + FL group appeared to be somewhat obscured (Table 2), suggesting that FL-induced mitophagy was concurrently suppressing gene expression. Furthermore, the HFD + FL group showed a more pronounced increase in steatosis score than the HFD group (Figure 1d). In drug-induced NAFLD, it has been shown that the drugs contribute to the development of steatosis by inhibiting mitochondrial β-oxidation [46]. Therefore, it is plausible that the HFD-mediated steatosis observed in the HFD + FL group may have been further exacerbated by FL-mediated inhibition of β-oxidation, as FL has been reported to downregulate the expression of genes associated with mitochondrial β-oxidation [25]. Consistent with this, FL suppressed the HFD-induced increase in the expression of *Ppara*, a transcription factor known to induce the expression of enzymes involved in HFD-induced β-oxidation (Table 2).

AMBRA1, expressed on the membranes of damaged mitochondria, plays a vital role in inducing mitophagy [14]. Moreover, it has been reported that AMBRA1 is explicitly expressed in tumorigenic lesions in a two-stage rat liver carcinogenesis model [47]. The increase in hepatic precancerous lesions observed in the HFD group was concurrent with the increased proportion of clusters showing high levels of AMBRA1 (C4 in Figure 3b–d) that was mitophagy inhibition (C3 in Figure 3f–h). However, these effects of HFD were counteracted upon co-administration with FL. Because the cluster showing Parkin-independent mitophagy induction was marginal in both the BD + FL and HFD + FL groups (C3 in Figure 3j–l), we could not definitively conclude that FL-induced mitophagy was Parkin-independent in hepatic preneoplastic lesions in the present study. Mitophagy generally suppresses carcinogenesis in the early stages, when normal cells transition into transformed cells [48]. In contrast, during late carcinogenesis, when transformed cells progress towards further malignant transformation and metastasis, mitophagy enhances carcinogenesis by eliminating damaged mitochondria, preventing ROS production, and enhancing mitochondrial energy production [48,49]. In this study, we observed precancerous lesions, which represent the early stage of carcinogenesis; therefore, it is possible that mitophagy inhibits the development of precancerous lesions, and that the HFD-induced inhibition of mitophagy increases the formation of hepatic precancerous lesions. This hypothesis is supported by the increased expression of LC3 and p62 that were previously observed in hepatic precancerous lesions [29,31]. Despite the marginal cumulative effects of FL and HFD on the generation of hepatic precancerous lesions (Figure 1h–j), it is conceivable that FL-induced mitophagy could be involved in the maintenance of certain populations of precancerous lesions in this group. These results suggest that mitophagy induction might lead to further malignant transformation of tumor cells. It is important to consider that counteracting factors such as AMBRA1 play an essential role in the process of hepatocarcinogenesis. This result is consistent with an increase in the formation of tumorigenic lesions when the autophagy inducer amiodarone was applied to a two-stage rat liver carcinogenesis model [47]. In addition to regulating mitophagy, *Ambra1* acts as a tumor suppressor gene in vivo; it inhibits cell cycle progression into the G1-S phase by promoting the degradation of cyclin D [50]. It interacts with protein phosphatase 2A (PP2A), a cell cycle regulator independent of autophagy regulation, and modulates cell growth and tumorigenesis by promoting the PP2A-mediated dephosphorylation of the oncogene *C-MYC* [51]. Notably, mutations in *Ambra1* have also been identified in various human tumors [50].

In conclusion, our study revealed that mitophagy was suppressed in the HFD group, while FL administration reversed this suppression, leading to an enhanced formation of precancerous lesions. Similar findings were also observed in the background hepatocytes other than the precancerous lesions, indicating a broader impact of mitophagy regulation beyond preneoplastic lesions. While current research focuses on developing mitophagy-inducing drugs for the treatment of NAFLD and cancer [52,53,54,55], it is crucial to consider the potential risk of mitophagy inducers aggravating NAFLD under a certain condition. By analyzing mitophagy both inside and outside precancerous lesions, we have provided a clear understanding of its pathological significance in NAFLD-associated carcinogenesis. As shown in the present study, employing cluster analysis using the autophagy marker LC3 and mitophagy indicators AMBRA1 and Parkin proves to be a valuable approach for studying mitophagy. However, a limitation of this study was that we did not demonstrate mitophagy in preneoplastic lesions with TEM. While TEM is a powerful tool, the weakness of this method is that only localized areas could be observed compared to broader histopathological and immunohistochemical examinations. Furthermore, in vitro and in vivo studies using primary hepatocytes and organoids are required to understand the role of FL on mitochondria toxicity and mitophagy. Our methodology can also be extended to the analysis of autophagy flux in preneoplastic lesions and micro-cancers, enabling a deeper exploration of the roles of mitophagy in malignant transformation and the development of novel therapeutic strategies preventing premalignant cells.

## 4. Methods

### 4.1. Chemicals

N-Nitrosodiethylamine (DEN; CAS No. 55-18-5, purity > 99%) and flutamide (FL; CAS No. 13311-84-7, purity > 98%) were purchased from Tokyo Kasei Kogyo (Chuo-ku, Tokyo, Japan) and Combi-Blocks (San Diego, CA, USA), respectively.

### 4.2. Animals and Treatments

A total of 30 5-week-old male F344/DuCrlCrlj rats were purchased from Charles River Laboratories Japan (Atsugi, Kanagawa, Japan) and reared under the following conditions: temperature 23 ± 3 °C, humidity 50 ± 20%, and 12 h light/12 h dark lighting cycle. The rats were fed a powdered diet (MF; Oriental Yeast Co., Itabashi-ku, Tokyo, Japan) as the basal diet (BD) and provided with drinking water in a clean rack equipped with rat paper-type enrichment, with a maximum of three animals per cage. In a previous study, rats were treated with 1000 or 2000 ppm doses of FL in feeding, resulting in final body weights of 85% and 81%, respectively, at the end of the study (8 weeks), compared to the control group [27]. Preliminary analyses were conducted on six animals, wherein FL was administered at three different doses (250, 500, and 1000 ppm in feeding) over a span of two weeks. Based on evaluations of body and liver weights, the FL dose of 500 ppm in feeding was selected for the present study. Six days before the start of the study, 12 rats were fed an HFD (D12451, 45 kcal%; Research Diets Inc., New Brunswick, NJ, USA), while the remaining rats were provided with a BD. This study was conducted as previously reported [28,29,30,31], according to an initiation and promotion model [32]. The rats were divided into four groups: BD-fed (BD, *n* = 6), BD-fed and FL-administered (BD + FL, *n* = 6), HFD-fed (HFD, *n* = 6), and HFD-fed and FL-administered (HFD + FL, *n* = 6). Throughout the study period, daily observations were performed to assess the general conditions of the animals. Additionally, the body weight and food and water intake measurements were performed weekly. At 13 weeks after study initiation, all animals were subjected to laparotomies under isoflurane anesthesia and humane euthanization through blood release. Subsequently, the liver and intraperitoneal fat, testis, epididymides, seminal vesicles/coagulating glands, and prostate were collected for organ weight measurement. The livers were fixed in 4% paraformaldehyde (PFA) for histopathological examination and immunohistochemical staining. A portion of the liver was frozen in liquid nitrogen and stored at −80 °C. The experimental plan involving animals was approved and reviewed by the Laboratory Animal Committee of Tokyo University of Agriculture and Technology (No. R03-149) prior to the commencement of the experiments. All animals were handled in accordance with the Guidelines for Animal Experimentation issued by the Japanese Association for Laboratory Animal Science (https://www.jalas.jp/english/en_about.html) (accessed on 10 April 2021). This study was conducted in compliance with the ARRIVE (Animal Research: Reporting of In Vivo Experiments) guidelines.

### 4.3. Histopathology

Following fixation in 4% PFA, liver specimens were subjected to thin sectioning to a thickness of approximately 3 μm. These sections were subsequently stained using hematoxylin and eosin (H&E) staining. In the liver tissue, we evaluated the NAFLD activity score (NAS), encompassing assessments from steatosis, ballooning, and inflammatory foci [30,56].

### 4.4. Immunohistochemistry and Clustering Analysis

The PFA-fixed liver specimens were thinly sliced and subjected to immunohistochemistry for targeting specific markers, including glutathione S-transferase placental form (GST-P), which is positive for rat liver proliferative lesions; Ki-67, a marker of cell proliferative activity; LC3, an autophagy marker; and AMBRA1 and Parkin, both pivotal regulators of mitophagy. Detailed information about the antibodies used, antigen activation methods, and antibody dilution ratios are presented in Appendix A, according to a previously reported method [57]. The number and area of the GST-P-positive foci were measured as previously described [28,32,57]. The positive rates of LC3, AMBRA1, and Parkin within each GST-P-positive focus were measured using Fiji (2.16.0, https://imagej.net/software/fiji/downloads (accessed on 15 April 2022)) (Appendix A). The number of Ki-67-positive cells per GST-P-positive focus was defined as the labeling index (%) [28,29,30,31]. GST-P-positive foci were randomly selected, and Ki-67-positive and negative cells inside the foci were counted to achieve a total of more than 1000 cells.

The GST-P-positive foci and the positivity rates for AMBRA1, LC3, and Parkin were collated for each individual and analyzed in groups, rather than individually, as previously reported [34]. For further analysis, the positive rates of LC3, AMBRA1, or Pink in GST-P-positive foci were plotted on the x- and y-axes, respectively. Subsequently, clustering was performed for analyzing each cluster using the *k*-means method. The positive rates of LC3, AMBRA1, and Parkin in background hepatocytes per four randomly selected fields at 400× magnification were also analyzed using Fiji. The AMBRA1, LC3, and Parkin positivity rates for each individual were tabulated and analyzed as a group rather than individually (a total of 96), as shown in the positive rates in GST-P-positive foci.

### 4.5. Ultrastructural Examination

TEM was conducted in background hepatocytes as previously reported [57]. Liver pieces of approximately 1 mm^3^ fixed in 4%PFA were further fixed in 2.5% glutaraldehyde with post-fixation with 1% osmic acid (Nisshin EM, #300, Shinjuku-ku, Tokyo, Japan), embedded in epoxy resin (TAAB, #T024, Aldermaston, Berkshire, UK), ultra-sectioned to 80 nm, stained with EM stainer (Nisshin EM, #336) and lead, and observed with JEM-1400Flash (JEOL Ltd., Akishima, Tokyo, Japan). TEM analysis was performed at Tokyo University of Agriculture and Technology for Smart-Core-Facility Promotion Organization.

### 4.6. Real-Time Reverse Transcription–Polymerase Chain Reaction Analysis

Expression analysis at the mRNA level was performed using the genes and primers listed in Appendix A. RT-PCR was performed as described previously [28] (see Appendix A for details).

### 4.7. Statistical Analyses

Means and standard deviations were calculated for all data. Statistical analyses were performed using either the Tukey or Steel-Dwass test. A significant level of 5% or less indicated a significant difference. A test of other proportions (Tukey–Kramer test) was performed to determine the proportion of clusters in each group.

## Figures and Tables

**Figure 1 ijms-26-02709-f001:**
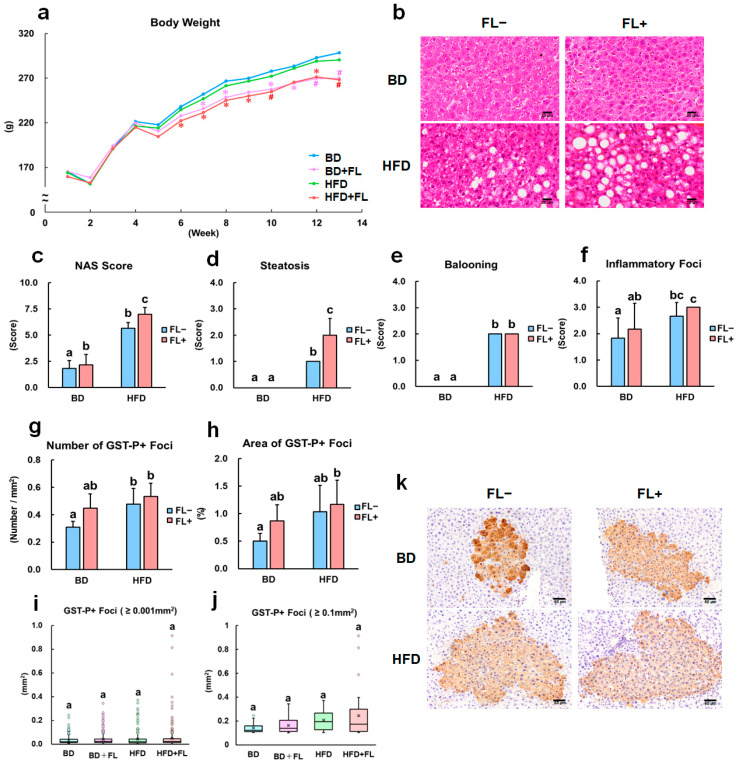
Body weight change, NAFLD activity score, and quantitative analysis of preneoplastic lesions in rats. Rats were initially fed a basal diet (BD) or high-fat diet (HFD) at the beginning of the study, with flutamide (FL) treatment initiated after 2 weeks. The rats were subjected to partial hepatectomy at 3 weeks and sacrificed at 13 weeks. (**a**) Body weight changes in the BD, BD + FL, HFD, and HFD + FL groups during the study. * *p* < 0.05 compared to the BD group, # *p* < 0.05 compared to the BD and HFD groups (Tukey’s or Steel–Dwass multiple comparison test). (**b**) Representative images illustrating steatosis in the HFD and HFD + FL groups. (**c**) NAFLD activity score (NAS). Scores for (**d**) steatosis, (**e**) ballooning, and (**f**) inflammatory foci. Quantitative analysis of the (**g**) number and (**h**) area of GST-P-positive foci. (**i**) Box plot depicting GST-P-positive foci larger than 0.001 mm^2^. (**j**) Box plot illustrating GST-P-positive foci larger than 0.1 mm^2^. (**k**) Representative images of GST-P foci. (**c**–**j**) Different letters indicate significant intergroup differences (*p* < 0.05, Tukey’s or Steel–Dwass test). Abbreviations: BD = basal diet; FL = flutamide; GST-P = glutathione S-transferase placental form; and HFD = high-fat diet.

**Figure 2 ijms-26-02709-f002:**
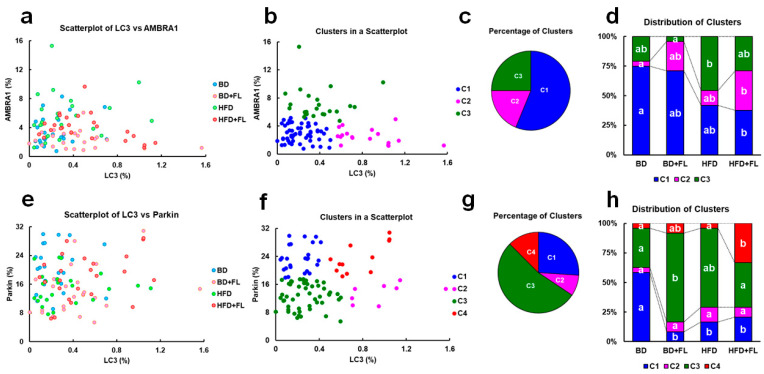
Scatterplot and clustering analysis of AMBRA1, Parkin, and LC3 expression in background hepatocytes (N = 96). Comparison of AMBRA1 and LC3 expression (%) (**a**–**d**) and of Parkin and LC3 (**e**–**h**) expression in the BD, BD + FL, HFD, and HFD + FL groups. Scatterplot (**a**,**e**), clustering analysis (**b**,**f**), percentage distribution of clusters (**c**,**g**), and distribution of clusters (**d**,**h**) in the BD, BD + FL, HFD, and HFD + FL groups (N = 24 in each group). (**b**) Clustering analysis of AMBRA1 and LC3 was conducted using *k*-means analysis (*k* = 3), and clusters C1, C2, and C3 were identified as representing basal level, mitophagy induction, and mitophagy inhibition, respectively, depending on the expression levels of each marker (see Appendix A). (**f**) Clustering analysis of Parkin and LC3 was conducted using *k*-means analysis (*k* = 4), and clusters CS1, C2, C3, and C4 were identified as representative of mitophagy inhibition, Parkin-independent mitophagy induction, basal level, and Parkin-dependent mitophagy induction, respectively, depending on expression levels of each marker (see Appendix A). (**d**,**h**) Bar chart showing the percentage distribution of clusters in each group. Different letters indicate significant intergroup differences (*p* < 0.05, significantly different by Tukey–Kramer test) (**d**,**h**). Abbreviations: AMBRA1 = autophagy and Beclin 1 regulator 1; BD = basal diet; FL = flutamide; HFD = high-fat diet; LC3, microtubule-associated protein 1A/1B-light chain 3.

**Figure 3 ijms-26-02709-f003:**
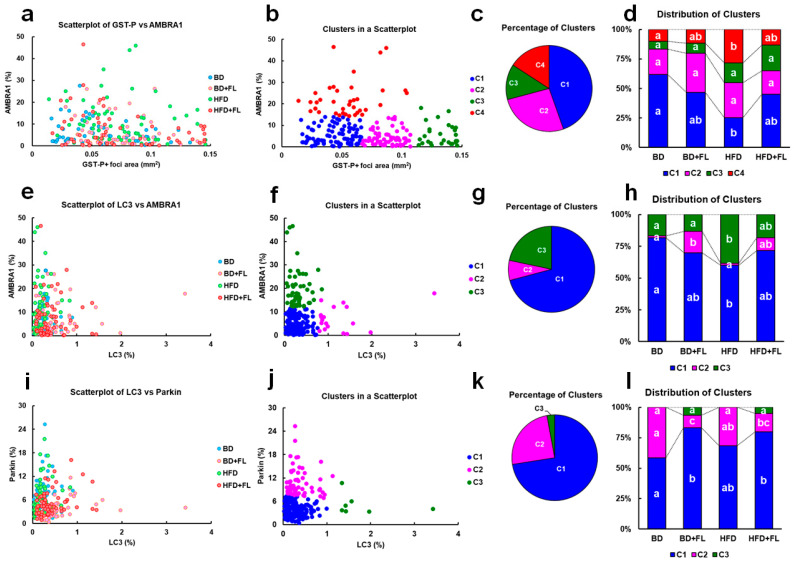
Scatterplot and clustering analysis of individual GST-P-positive focus areas and AMBRA1, Parkin, and LC3 expression rates in hepatic preneoplastic lesions (N = 240). Comparison of the areas of individual GST-P-positive foci and AMBRA1 expression (%) (**a**–**d**), LC3 and AMBRA1 expression (%) (**e**–**h**), and LC3 and Parkin expression (%) (**i**–**l**). Scatterplot (**a**,**e**,**i**), clustering analysis (**b**,**f**,**j**), percentage distribution of clusters (**c**,**g**,**k**), and distribution of clusters (**d**,**h**,**l**) in the BD, BD + FL, HFD, and HFD + FL groups (N = 60 in each group). (**b**,**f**,**j**) Clustering analysis was conducted using *k*-means analysis (*k* = 3 or 4). (**b**) Clustering analysis of GST-P-positive foci and AMBRA1 reveals that clusters C1, C2, C3, and C4 represent small-, middle-, and large-sized foci, and high level of AMBRA1 (mitophagy inhibition), respectively, depending on expression levels of each marker (see Appendix A). (**f**) Clustering analysis of LC3 and AMBRA1 reveals that clusters C1, C2, and C3 represent basal level, mitophagy induction, and mitophagy inhibition, respectively, depending on expression levels of each marker (see Appendix A). (**j**) Clustering analysis of LC3 and Parkin reveals that clusters C1, C2, and C3 represent basal level, mitophagy inhibition, and Parkin-independent mitophagy induction, respectively, depending on expression levels of each marker (see Appendix A). (**d**,**h**,**l**) Bar chart showing the percentage distribution of clusters in each group. Different letters indicate significant intergroup differences (*p* < 0.05, significantly different by Tukey–Kramer test) (**d**,**h**,**l**). Abbreviations: AMBRA1 = autophagy and Beclin 1 regulator 1; BD = basal diet, FL = flutamide, GST-P = glutathione S-transferase placental form, HFD = high-fat diet; LC3, microtubule-associated protein 1A/1B-light chain 3.

**Table 1 ijms-26-02709-t001:** Final body weight, organ weight, food intake, and water intake in rats.

Group	BD	BD + FL	HFD	HFD + FL
No. of animals	6	6	6	6
Final body weight (g)	301.5 ± 15.15 ^a^	273.7 ± 6.96 ^bc^	293.4 ± 15.13 ^ab^	271.8 ± 12.05 ^c^
Food intake (g/kg)	47.1 ± 16.8 ^a^	46.0 ± 15.5 ^a^	34.1 ± 15.0 ^a^	33.3 ± 13.9 ^a^
Water intake (g/kg)	66.6 ± 19.3 ^a^	64.4 ± 17.3 ^a^	53.8 ± 17.6 ^a^	51.3 ± 16.5 ^a^
Absolute liver weight (g)	7.68 ± 0.72 ^ab^	7.91 ± 0.32 ^a^	6.89 ± 0.86 ^b^	6.90 ± 0.38 ^b^
Relative liver weight (%BW)	2.54 ± 0.13 ^a^	2.89 ± 0.07 ^b^	2.34 ± 0.19 ^a^	2.54 ± 0.16 ^a^
Absolute intraperitoneal fat weight (g)	6.27 ± 1.47 ^a^	4.40 ± 0.40 ^b^	6.78 ± 0.86 ^a^	5.04 ± 1.33 ^ab^
Relative intraperitoneal fat weight (%BW)	2.07 ± 0.46 ^ab^	1.61 ± 0.13 ^b^	2.30 ± 0.18 ^a^	1.85 ± 0.46 ^ab^
Absolute testis weight (g)	3.00 ± 0.07 ^a^	2.77 ± 0.14 ^a^	2.94 ± 0.15 ^a^	2.94 ± 0.18 ^a^
Relative testis weight (%BW)	1.00 ± 0.04 ^a^	1.01 ± 0.04 ^a^	1.00 ± 0.03 ^a^	1.08 ± 0.07 ^a^
Absolute epididymis weight (g)	1.04 ± 0.06 ^a^	0.48 ± 0.09 ^c^	0.90 ± 0.09 ^b^	0.56 ± 0.08 ^c^
Relative epididymis weight (%BW)	0.34 ± 0.02 ^a^	0.17 ± 0.03 ^b^	0.31 ± 0.03 ^a^	0.21 ± 0.04 ^b^
Absolute SC/CG weight (g)	1.16 ± 0.18 ^a^	0.18 ± 0.05 ^b^	1.05 ± 0.13 ^a^	0.24 ± 0.08 ^b^
Relative SC/CG weight (%BW)	0.39 ± 0.07 ^a^	0.07 ± 0.02 ^b^	0.36 ± 0.03 ^a^	0.09 ± 0.03 ^ab^
Absolute prostate weight (g)	0.37 ± 0.03 ^a^	0.07 ± 0.03 ^b^	0.33 ± 0.07 ^a^	0.07 ± 0.04 ^b^
Relative prostate weight (%BW)	0.12 ± 0.01 ^a^	0.03 ± 0.01 ^b^	0.11 ± 0.02 ^a^	0.03 ± 0.01 ^b^

Abbreviations: BD, basal diet; SC/CG, seminal vesicle/coagulating gland; HFD, high-fat diet; FL, flutamide, BW; body weight. Data are shown as the mean ± standard deviation. Different letters indicate significant differences between groups (*p* < 0.05, significantly different by Tukey’s or Steel-Dwass test).

**Table 2 ijms-26-02709-t002:** Gene expression analysis of autophagy, mitochondria/mitophagy, lipid metabolism, drug metabolism, inflammation, and oxidative stress in liver samples.

Group		BD	BD + FL	HFD	HFD + FL
No. of animals		6	6	6	6
Autophagy-related genes					
	*Atg3*	1.05 ± 0.34 ^a^	1.69 ± 0.40 ^a^	2.58 ± 1.23 ^a^	1.60 ± 0.32 ^a^
	*Atg5*	1.10 ± 0.51 ^a^	1.77 ± 0.72 ^ab^	2.61 ± 0.91 ^b^	2.27 ± 0.83 ^ab^
	*Atg7*	1.03 ± 0.26 ^a^	1.10 ± 0.23 ^a^	2.08 ± 0.64 ^b^	1.98 ± 1.02 ^ab^
	*Lamp1*	1.06 ± 0.40 ^a^	1.50 ± 0.41 ^ab^	2.08 ± 0.72 ^b^	1.54 ± 0.59 ^ab^
	*Lamp2*	1.05 ± 0.35 ^a^	1.57 ± 0.36 ^ab^	2.13 ± 0.93 ^b^	1.19 ± 0.39 ^a^
	*Lc3*	1.00 ± 0.08 ^a^	1.24 ± 0.38 ^ab^	2.06 ± 0.83 ^b^	1.58 ± 1.04 ^ab^
	*p62*	1.03 ± 0.26 ^a^	1.30 ± 0.57 ^a^	1.82 ± 0.72 ^a^	1.67 ± 1.02 ^a^
Mitochondria/Mitophagy-related genes					
	*Ambta1*	1.14 ± 0.66 ^a^	1.98 ± 0.75 ^a^	3.95 ± 3.18 ^a^	4.64 ± 5.32 ^a^
	*Parkin*	1.17 ± 0.64 ^a^	2.54 ± 0.99 ^ab^	3.27 ± 2.12 ^ab^	3.57 ± 1.59 ^b^
	*Pink1*	1.03 ± 0.24 ^a^	1.09 ± 0.22 ^a^	1.81 ± 0.65 ^a^	1.12 ± 0.29 ^a^
	*Bnip3*	1.01 ± 0.18 ^a^	1.48 ± 0.41 ^a^	1.43 ± 0.47 ^a^	1.05 ± 0.34 ^a^
	*Nadh*	1.03 ± 0.26 ^a^	1.32 ± 0.26 ^a^	2.21 ± 0.97 ^a^	1.17 ± 0.32 ^a^
	*Sdhd*	1.04 ± 0.30 ^a^	1.67 ± 0.40 ^ab^	2.17 ± 0.86 ^b^	1.51 ± 0.51 ^ab^
	*ATP synthase*	1.03 ± 0.29 ^a^	1.47 ± 0.45 ^a^	1.79 ± 0.75 ^a^	1.24 ± 0.30 ^a^
Lipid metabolism-related genes					
	*Abca1*	1.09 ± 0.40 ^a^	1.89 ± 0.79 ^ab^	3.26 ± 1.07 ^c^	2.74 ± 0.89 ^bc^
	*Acox1*	1.14 ± 0.59 ^a^	1.80 ± 0.54 ^a^	2.74 ± 1.36 ^a^	1.89 ± 0.34 ^a^
	*Apob*	1.05 ± 0.30 ^a^	0.82 ± 0.21 ^a^	1.49 ± 0.76 ^a^	1.31 ± 0.32 ^a^
	*Dgat2*	1.06 ± 0.39 ^a^	1.36 ± 0.41 ^ab^	1.77 ± 0.51 ^b^	1.70 ± 0.35 ^ab^
	*Fasn*	1.27 ± 0.87 ^a^	3.04 ± 1.50 ^a^	1.72 ± 0.96 ^a^	2.43 ± 2.25 ^a^
	*Hmgcs1*	1.09 ± 0.47 ^a^	0.72 ± 0.36 ^a^	1.51 ± 0.77 ^a^	1.18 ± 0.36 ^a^
	*Hsd3 b1*	1.10 ± 0.48 ^a^	0.38 ± 0.40 ^a^	2.17 ± 0.90 ^b^	0.83 ± 0.60 ^a^
	*Lpl*	1.06 ± 0.37 ^a^	0.97 ± 0.27 ^a^	1.50 ± 0.80 ^a^	1.17 ± 0.46 ^a^
	*Lss*	1.11 ± 0.53 ^a^	1.46 ± 0.72 ^a^	1.69 ± 0.66 ^a^	2.30 ± 1.19 ^a^
	*Ppara*	1.19 ± 0.69 ^a^	1.81 ± 0.97 ^a^	4.55 ± 1.44 ^b^	2.08 ± 0.71 ^a^
	*Pparg*	1.05 ± 0.36 ^a^	1.16 ± 0.98 ^a^	0.61 ± 0.26 ^a^	1.91 ± 1.47 ^a^
	*Scd1*	1.03 ± 0.26 ^a^	1.27 ± 0.38 ^a^	1.69 ± 0.46 ^a^	1.25 ± 0.62 ^a^
	*Srebf1*	1.17 ± 0.73 ^a^	0.84 ± 0.43 ^a^	1.59 ± 1.26 ^a^	1.58 ± 1.02 ^a^
	*Srebf2*	1.08 ± 0.41 ^a^	1.53 ± 0.44 ^ab^	2.49 ± 0.95 ^b^	2.33 ± 0.88 ^b^
Drug metabolism-related genes					
	*Cyp1a1*	1.27 ± 1.09 ^a^	73.20 ± 45.16 ^b^	1.90 ± 0.94 ^a^	96.14 ± 93.51 ^b^
	*Cyp2b1*	1.64 ± 1.91 ^a^	1.97 ± 0.95 ^a^	1.84 ± 0.91 ^a^	2.51 ± 1.10 ^a^
	*Cyp3a1*	1.05 ± 0.34 ^a^	0.65 ± 0.25 ^a^	1.40 ± 0.54 ^a^	0.89 ± 0.79 ^a^
Inflammation related genes					
	*Tnf-a*	1.11 ± 0.51 ^a^	1.29 ± 0.64 ^a^	1.81 ± 1.11 ^a^	3.33 ± 3.87 ^a^
Oxidative stress-related genes					
	*Catalase*	1.04 ± 0.33 ^a^	1.75 ± 0.58 ^ab^	2.70 ± 1.22 ^b^	2.18 ± 0.98 ^ab^
	*Gpx1*	1.16 ± 0.70 ^a^	1.68 ± 0.67 ^a^	1.92 ± 0.75 ^a^	2.05 ± 0.95 ^a^
	*Gpx2*	1.08 ± 0.47 ^a^	5.36 ± 1.91 ^b^	3.44 ± 2.86 ^ab^	4.98 ± 2.16 ^b^
	*Mn-SOD*	1.04 ± 0.32 ^a^	1.23 ± 0.38 ^a^	1.65 ± 0.73 ^a^	1.54 ± 0.56 ^a^
	*Sod1*	1.11 ± 0.53 ^a^	1.32 ± 0.54 ^a^	1.18 ± 0.55 ^a^	1.00 ± 0.40 ^a^
	*Sod2*	1.13 ± 0.63 ^a^	1.41 ± 0.23 ^a^	2.07 ± 0.87 ^a^	1.33 ± 0.35 ^a^

Abbreviations: BD, basal diet; HFD, high-fat diet; FL, flutamide. Data are shown as the mean ± standard deviation. Different letters indicate significant differences between groups (*p* < 0.05, significantly different by Tukey’s or Steel-Dwass test).

## Data Availability

The data generated or analyzed during this study are provided in this published article and Appendix A.

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
