# Peer review of "Flutamide Promotes Early Hepatocarcinogenesis Through Mitophagy in High-Fat Diet-Fed Non-Obese Steatotic Rats"

_ijms, 2025, doi:10.3390/ijms26062709_

Round 1
Reviewer 1 Report
Comments and Suggestions for Authors
In this manuscript, the authors have shown the effect of flutamide in inducing mitophagy in a mice model of NAFLD. The authors suggested that mitophagy was reduced in NAFLD, and flutamide treatment in the NFALD group changes the hepatic gene expression. The study is interesting. My comments are provided below
1. The manuscript needs to be edited. The introduction and discussion section is very lengthy.
2. The authors showed the increased in mitophagy usin autophagy markers LC3 and AMBRA1 and Parkin. However, the authors didnot perform any functional test for mitophagy using primary hepatocytes cultured from these rats. The use of any mitophagy inhibitors would be more valuable.
3. The number of animal used per group is 6. Is there any specific reason for not using n=10 per group.
4. Did the authors analyze the level of testosterone in these rats after treating them with flutamide?
5. Did the author analyze the mitochondrial defect in the ultrastructural level in these rats?
6. Did the authors analyze the effect of flutamide on Complex I, III and V activity?
7. The changes in gene expression should be analyzed by western blot analyses.
Comments on the Quality of English Language
The English language needs to be improved. there are many spelling errors throughout the text. The sentences are lengthy.
Author Response
Comment 1.
The manuscript needs to be edited. The introduction and discussion section is very lengthy.
(Response) Thank you for your suggestion, we have revised the sections of Introduction and Discussion to be more focused and concise overall.
Comment 2.
The authors showed the increased in mitophagy using autophagy markers LC3 and AMBRA1 and Parkin. However, the authors did not perform any functional test for mitophagy using primary hepatocytes cultured from these rats. The use of any mitophagy inhibitors would be more valuable.
(Response) Thank you very much for your critical comments. In this study, we attempted to generate liver organoids from the in vivo samples, but unfortunately, we were not able to continue the culture long enough to be able to perform in vitro experiments. We understand that confirming the mitochondrial toxicity of flutamide under culture conditions and observing the subsequent mitophagy is a critical approach to ensure the results in vivo experiment. Furthermore, attempts to recapitulate NAFLD pathology under cellular conditions have also been extensively studied. It is very important approach to reproduce NAFLD conditions in vitro and to clarify the significance of mitophagy in flutamide and HFD treatment using mitophagy inhibitors. Throughout our project, we are attempting to establish liver organoids (see references below), and pursuing more detailed mitophagy mechanisms will be a critical future study.
Yoshida T, Kobayashi M, Uomoto S, Ohshima K, Hara E, Katoh Y, Takahashi N, Harada T, Usui T, Elbadawy M, Shibutani M. The potential of organoids in toxicologic pathology: role of toxicologic pathologists in in vitro chemical hepatotoxicity assessment. J Toxicol Pathol. 2022 Jul;35(3):225-235. doi: 10.1293/tox.2022-0017. Epub 2022 May 23. PMID: 35832897; PMCID: PMC9256002.
Elbadawy M, Yamanaka M, Goto Y, Hayashi K, Tsunedomi R, Hazama S, Nagano H, Yoshida T, Shibutani M, Ichikawa R, Nakahara J, Omatsu T, Mizutani T, Katayama Y, Shinohara Y, Abugomaa A, Kaneda M, Yamawaki H, Usui T, Sasaki K. Efficacy of primary liver organoid culture from different stages of non-alcoholic steatohepatitis (NASH) mouse model. Biomaterials. 2020 Apr;237:119823. doi: 10.1016/j.biomaterials.2020.119823. Epub 2020 Jan 27. PMID: 32044522.
Comment 3.
The number of animal used per group is 6. Is there any specific reason for not using n=10 per group.
(Response) Thank you for your suggestion. In the past, my laboratory has used more than 10 cases per group when conducting experiments using this model (Murayama et al., 2018 Masuda et al., 2019; Nakamura et al., 2018; Eguchi et al., 2022). I had worked in a toxicology laboratory in a previous job, so I am familiar with the use of N=10 in general toxicology studies. However, we chose N=6 based on the 3R concept of animal welfare and also to allow for statistical evaluation of body weight, organ weights, pathological examination, etc.
We understand that the common method of evaluating precancerous lesions is to calculate the average value for each individual and then calculate the average value for the group. However, the variability of the analyzed data in precancerous lesions is an important issue to be solved, and we thought that increasing the number of N would not completely solve this problem. Further analysis with more animals might, in some cases, alleviate the problem. However, that choice is an approach relative to the 3R concept of animal welfare, and our research group decided not to make that choice. As for the analysis of precancerous lesions, considering their diversity and as a method that allows evaluation with this number of animals, we succeeded in introducing a method to aggregate all individual precancerous lesions within a group and evaluate them while considering the diversity of precancerous lesions (Uomoto et al., 2023). We adopted this method in this study as well.
Murayama, H., Eguchi A, Nakamura M, Kawashima M, Nagahara R, Mizukami S, Kimura M, Makino E, Takahashi N, Ohtsuka R, Koyanagi M, Hayashi SM, Maronpot RR, Shibutani M, Yoshida T. Spironolactone in combination with α-glycosyl isoquercitrin prevents steatosis-related early hepatocarcinogenesis in rats through the Observed NADPH Oxidase Modulation. Toxicol. Pathol. 2018, 46, 530–539. doi: 10.1177/0192623318778508.
Masuda, S., Mizukami S, Eguchi A, Ichikawa R, Nakamura M, Nakamura K, Okada R, Tanaka T, Shibutani M, Yoshida T. Immunohistochemical expression of autophagosome markers LC3 and p62 in preneoplastic liver foci in high fat diet-fed rats. J. Toxicol. Sci. 2019, 44, 565–574. doi: 10.2131/jts.44.565.
Nakamura M., Eguchi A, Inohana M, Nagahara R, Murayama H, Kawashima M, Mizukami S, Koyanagi M, Hayashi SM, Maronpot RR, Shibutani M, Yoshida T. Differential impacts of mineralocorticoid receptor antagonist potassium canrenoate on liver and renal changes in high fat diet-mediated early hepatocarcinogenesis model rats. J. Toxicol. Sci. 2018, 43, 611–621. doi: 10.2131/jts.43.611.
Eguchi, A., Mizukami S, Nakamura M, Masuda S, Murayama H, Kawashima M, Inohana M, Nagahara R, Kobayashi M, Yamashita R, Uomoto S, Makino E, Ohtsuka R, Takahashi N, Hayashi SM, Maronpot RR, Shibutani M, Yoshida T. Metronidazole enhances steatosis-related early-stage hepatocarcinogenesis in high fat diet-fed rats through DNA double-strand breaks and modulation of autophagy. Environ. Sci. Pollut. Res. Int. 2022, 29, 779–789. doi: 10.1007/s11356-021-15689-2.
Uomoto, S., Takesue K, Shimizu S, Maeda N, Oshima K, Hara E, Kobayashi M, Takahashi Y, Shibutani M, Yoshida T. Phenobarbital, a hepatic metabolic enzyme inducer, inhibits preneoplastic hepatic lesions with expression of selective autophagy receptor p62 and ER-phagy receptor FAM134B in high-fat diet-fed rats through the inhibition of ER stress. Food Chem. Toxicol. 2023, 173, 113607. doi: 10.1016/j.fct.2023.113607.
Comment 4
Did the authors analyze the level of testosterone in these rats after treating them with flutamide?
(Response) Thank you for your comments. We did not measure serum testosterone in this study. However, we did measure weights of testis and accessory gland, so we have added these weight data to Table 1, which shows that FL administration did not result in testicular weights, but did decrease epididymis, seminal vesicle/coagulation gland, and prostate weights, as described on page 6, L104-107; page 14, L245 and 246. The results showed that FL had a sufficient antiandrogenic effect under the conditions of this study.
Comment5.
Did the author analyze the mitochondrial defect in the ultrastructural level in these rats?
(Response) Thank you for your very important point. In this study, we were not able to observe mitochondrial loss in several TEM. We should consider that FL might induce mitophagy, following loss of mitochondria due to mitochondrial toxicity in a time-course study. We would like to continue our research.
Comment 6.
Did the authors analyze the effect of flutamide on Complex I, III and V activity?
(Response) Thank you for your comments. We have not measured the activity of these enzymes in this study. We had measured them by gene expression analysis on page 12, L204-215. We would like to include the experiments in our future studies.
Comment 7.
The changes in gene expression should be analyzed by western blot analyses.
(Response) Thank you for your suggestion. It would be a very important additional study to confirm the results of the gene expression analysis with western blots. In this study, we observed differences in mitophagy between precancerous lesions and background hepatocytes, and we believe that homogenizing liver tissue for enzyme activity and protein expression is not suitable for the purpose of the experiment. We would like to consider these in future additional studies, including in vitro experiments.
Comments on the Quality of English Language:
The English language needs to be improved. there are many spelling errors throughout the text. The sentences are lengthy.
(Response) Thank you for pointing this out. We apologize for the many typographical errors and other errors in the text. We have reviewed the entire document and corrected them.
Reviewer 2 Report
Comments and Suggestions for Authors
The authors of this study investigated the effects of non-alcoholic fatty liver disease (NAFLD) and hepatocarcinogenicity of a combination of flutamide (FL) and a high-fat diet (HDF). The combination of FL and HDF significantly enhanced some aspects of NAFLD in the rat liver. However, it did not enhance the carcinogenic effects detected with Placental glutathione S-transferase (GST-P)-positive foci. Furthermore, this study demonstrated that FT and HFD have opposing effects on mitochondria. They presented some impressive data, but there were some questions ad suggestions as described below.
Major comments
- In Figure S2, the images depicting autophagy and Beclin 1 regulator 1 (AMBRA1) staining do not align with the graph data. Specifically, the staining in the BD+FL and HFD+FL groups looks lower compared to the BD group in Figure S2c. Furthermore, the staining area in the foci is lower than that observed in the background hepatocytes (Figure S2a and c), although the positive percentage in the foci is higher (Figure S2a and c). The authors should recheck the data.
- In Figure 2, the authors used a total of 96 background hepatocyte areas. Because they only had six rats per group, they need to detail the number of 96 areas. Did they select and examine four areas per individual rat in this study?
- To focus on the changes in AMBRA1 expression induced by HFD and FL, they each regulated expression in different directions in the liver. Meanwhile, the authors considered that “In the HFD+FL group, the downregulation of AMBRA1 may potentially contribute to the formation of precancerous lesions” on page 19, line 340 to 342. Given the lack of a significant difference in the carcinogenic effects of HDF and HDF+FL, it is pertinent to ask whether factors whose effects are offset, such as AMBRA1, could play a significant role in the process of hepatocarcinogenesis.
- Although the authors cite a paper, they do not describe real-time reverse transcription-polymerase chain reaction analysis. Therefore, the used products, including company name, internal control gene information, and expression analysis method, are unclear. In particular, the internal control is not listed in Table S6, which makes it impossible to confirm reproducibility, which is a problem.
Minor comments
- The authors should write the formal name of “GST-P” on page 7, line 114.
- The authors wrote the sentence, “We analyzed autophagy-related genes and showed that the expression of Atg5, Atg7, Lamp1, Lamp2, and Lc3 was significantly increased in the HFD group compared to the BD group. However, when HFD feeding was combined with FL administration, a significant decrease was observed in the expression of these genes (Table 2).” on page 12, lines 200 to 203, but the significantly decrease of Atg5, Atg7, Lamp1, and Lc3 in HFD+FL group was not detected. They should correct them.
- How did the authors administer FL to the rats in this study on page 21, lines 370 to 374? By feeding them? They should write that.
- The reference style of “32” on page 21, line 377 should be corrected.
- How to check the number of Ki-67-positive cells per GST-P-positive focus on page 23, line 405? Reference 31 did not present this information clearly. The authors should write it.
- How to get the liver pieces for the transmission electron microscope (TEM) on page 23, line 412? The authors should write it in the animal experiment.
Author Response
Major comments
Comment1:
In Figure S2, the images depicting autophagy and Beclin 1 regulator 1 (AMBRA1) staining do not align with the graph data. Specifically, the staining in the BD+FL and HFD+FL groups looks lower compared to the BD group in Figure S2c. Furthermore, the staining area in the foci is lower than that observed in the background hepatocytes (Figure S2a and c), although the positive percentage in the foci is higher (Figure S2a and c). The authors should recheck the data.
(Response) Thank you for your suggestion. We have reviewed the data. The analysis data of AMBRA1 in the background hepatocytes and precancerous lesions are properly generated as Figures. We have added the analysis data for each rat to Figure S2 for reference. There are large individual differences in the analysis data of background hepatocytes, and some individuals with high values and some with low values are included. Indeed, the expression within foci tends to be lower compared to the surrounding hepatocytes, so we have selected such pictures. We believe that precancerous lesions are regulated by a specific niche for mitophagy, but the differences from mitophagy and mitochondria dynamics between background hepatocytes and preneoplastic lesions require further detailed molecular pathological analysis.
Comment 2: In Figure 2, the authors used a total of 96 background hepatocyte areas. Because they only had six rats per group, they need to detail the number of 96 areas. Did they select and examine four areas per individual rat in this study?
(Response) You are correct. We have included the details on page 22, lines 388-391, as follows: The positive rates of LC3, AMBRA1, and Parkin in background hepatocytes per four randomly selected fields at 400x magnification were also analyzed using Fiji. The AMBRA1, LC3, and Parkin positivity rates for each individual were tabulated and analyzed as a group rather than individually (a total of 96), as shown in the positive rates in GST-P-positive foci.
Comment 3: To focus on the changes in AMBRA1 expression induced by HFD and FL, they each regulated expression in different directions in the liver. Meanwhile, the authors considered that “In the HFD+FL group, the downregulation of AMBRA1 may potentially contribute to the formation of precancerous lesions” on page 19, line 340 to 342. Given the lack of a significant difference in the carcinogenic effects of HDF and HDF+FL, it is pertinent to ask whether factors whose effects are offset, such as AMBRA1, could play a significant role in the process of hepatocarcinogenesis. 
(Response) Thank you for your very valuable comments. We have included a discussion of these considerations in L 289-316 on pages 17-18.
Comment 4: Although the authors cite a paper, they do not describe real-time reverse transcription-polymerase chain reaction analysis. Therefore, the used products, including company name, internal control gene information, and expression analysis method, are unclear. In particular, the internal control is not listed in Table S6, which makes it impossible to confirm reproducibility, which is a problem.
(Response) Thank you for your comments. The detailed method is described in Supplemental Materials. The internal control, Hprt1 is listed in revised Table S6.
Minor comments
Comment 5: The authors should write the formal name of “GST-P” on page 7, line 114.
(Response) Thank you for pointing this out. We have corrected it according to your suggestion on pages 6 and 7, lines 108 and 109.
Comment 6: The authors wrote the sentence, “We analyzed autophagy-related genes and showed that the expression of Atg5, Atg7, Lamp1, Lamp2, and Lc3 was significantly increased in the HFD group compared to the BD group. However, when HFD feeding was combined with FL administration, a significant decrease was observed in the expression of these genes (Table 2).” on page 12, lines 200 to 203, but the significantly decrease of Atg5, Atg7, Lamp1, and Lc3 in HFD+FL group was not detected. They should correct them.
(Response) Thank you for pointing this out. We have corrected the following on Page 11, lines 197 and 198, as follows: . However, when HFD feeding was combined with FL administration, a significant decrease was not observed in the expression of these genes except for Lamp2 (Table 2).
Comment 7: How did the authors administer FL to the rats in this study on page 21, lines 370 to 374? By feeding them? They should write that.
(Answer) Thank you very much for your attention. We have added “in feeding” to each sentence in the previous report [27], the preliminary study and the main study, because FL is mixed with food and administered in the main study on page 20, lines 346-349.
Comment 8: The reference style of “32” on page 21, line 377 should be corrected.
(Answer) I am very sorry. We revised the sentence on page 20, line 353.
Comment 9: How to check the number of Ki-67-positive cells per GST-P-positive focus on page 23, line 405? Reference 31 did not present this information clearly. The authors should write it.
(Response) Thank you for pointing this out. We randomly selected GST-P positive foci and counted the positive and negative cells inside the positive foci so that the total number of positive and negative cells was greater than 1000 cells. We revised the sentence on page 24, lines 383 and 384, as follows: GST-P-positive foci were randomly selected, and Ki-67-positive and negative cells inside the foci were counted to achieve a total of more than 1,000 cells.
Comment 10: How to get the liver pieces for the transmission electron microscope (TEM) on page 23, line 412? The authors should write it in the animal experiment.
(Response) Thank you for your suggestion. For the electron microscope material, we used 4% PFA-fixed liver fragments. These pieces were further fixed in 2.5% glutaraldehyde with post-fixation with 1% osmic acid. We revised the sentence on page 22, line 394.
“Liver pieces of approximately 1 mm3 had fixed in 4%PFA were further fixed in 2.5% glutaraldehyde with post-fixation with 1% osmic acid”
Round 2
Reviewer 1 Report
Comments and Suggestions for Authors
The authors have added some additional data in the revised manuscript. Further, the authors included the limitation of the study in the discussion section. Due to the inherent nature of the organoid culture difficulties, the authors couldnot perform the some of the molecular aspect of flutamide. The study is novel and interesting. Overall, the revised version of the manuscript is improved a lot. I support the publication of the revised manuscript.
Reviewer 2 Report
Comments and Suggestions for Authors
The authors have improved the manuscript with the reviewer's comments. However, one minor issue remains.
Minor comment
In the response for the major comment, the authors wrote that "the expression within foci tends to be lower compared to the surrounding hepatocytes" However, the graph shows that the percentage of AMBRA1 within foci (Figure S2d) is higher than that within non-foci (Figure S2b). The authors need to explain the discrepancy between them.